# Sulfonated PAM/PPy Cryogels with Lowered Evaporation Enthalpy for Highly Efficient Photothermal Water Evaporation

**DOI:** 10.3390/polym15092108

**Published:** 2023-04-28

**Authors:** Shi-Chang Hou, Dao-Wei Zhang, Jun Chen, Xiao-Xiao Guo, Abdul Haleem, Wei-Dong He

**Affiliations:** 1CAS Key Laboratory of Soft Matter Chemistry, Department of Polymer Science and Engineering, University of Science and Technology of China, Hefei 230026, China; 2School of Chemistry and Chemical Engineering, Jiangsu University, Zhenjiang 212013, China

**Keywords:** polymeric cryogel, polyacrylamide, photothermal evaporation, evaporation enthalpy, sulfonating modification

## Abstract

Because of the increasing scarcity of water resources, the desalination of seawater by photothermal evaporation with harvested solar energy has gradually become a popular research topic. The interconnected macroporous cryogel prepared from polymerization and crosslinking below the freezing temperature of the reactant solution has an excellent performance in photothermal water evaporation after loading photothermal materials. In this study, polyacrylamide (PAM) cryogels were prepared by cryo-polymerization and sulfonated in an alkaline solution containing formaldehyde and Na_2_SO_3_. Importantly, the evaporation enthalpy of water in sulfonated PAM cryogel was reduced to 1187 J·g^−1^ due to the introduction of sulfonate groups into PAM, which was beneficial to increase the photothermal evaporation rate and efficiency. The sulfonated PAM cryogels loaded with polypyrrole and the umbrella-shaped melamine foam substrate were combined to form a photothermal evaporation device, and the evaporation rate was as high as 2.50 kg·m^−2^·h^−1^ under one-sun radiation. Meanwhile, the evaporation rate reached 2.09 kg·m^−2^·h^−1^ in the 14 wt% high-concentration saline solution, and no salt crystals appeared on the surface of the cryogel after 5 h of photothermal evaporation. Therefore, it was evidenced that the presence of sulfonate groups not only reduced the evaporation enthalpy of water but also prevented salting-out from blocking the water delivery channel during photothermal evaporation, with a sufficiently high evaporation rate, providing a reliable idea of matrix modification for the design of high-efficiency photothermal evaporation materials.

## 1. Introduction

The shortage of fresh water has gradually become a serious challenge with the continuous development of modern society and the increasing population [1,2], and desalination has long been considered as a potential solution to the growing water scarcity problem [3,4]. With the development of solar energy collection and utilization technology, one research goal of new water purification technology is solar evaporation, which can directly use sunlight as a sustainable energy source [5,6]. Solar evaporation, as one of the most widely studied research areas, has attracted more and more attention in the past decade due to its low energy consumption, high efficiency, and little impact on the environment [7,8,9]. The performance evaluation of solar photothermal evaporation is mainly based on its evaporation rate and efficiency, and is greatly affected by the photothermal conversion efficiency, rapid water transport, evaporation enthalpy of water, and so on [10,11]. At present, various photothermal materials based on carbon (graphene [12] and carbon nanotubes [13], etc.), metal nanoparticles (gold, platinum, etc.) [14,15], conjugated polymers (polypyrrole and polyaniline, etc.) [16,17,18,19], and semiconductors [20] have been developed and combined with hydrophilic matrix materials to achieve solar vapor generation with high evaporation efficiency [21,22]. However, the hydrophobicity of most photothermal materials frequently makes the evaporation rate of solar steam power generation lower than 2 kg·m^−2^·h^−1^ under one-sun radiation [23,24,25], and the change in the hydrophilicity of the photothermal evaporation device greatly increases its evaporation rate [26,27,28].

So far, the photothermal evaporation rate has been greatly improved through the intensive efforts of researchers to reduce the evaporation enthalpy of water [29,30], which is achieved by changing the hydrophilicity of the photothermal evaporation device. Bai et al. [31] prepared double N, O-doped carbon foam (NCF) from poly(ethylene terephthalate) waste with low-temperature carbonization at 340 °C. Due to the superhydrophilicity of rich N, O-containing groups, NCF promoted the formation of water clusters and reduced the enthalpy of water evaporation by about 37%, so that NCF exhibited a high evaporation rate (2.4 kg·m^−2^·h^−1^). Nabeela et al. [32] reported a self-floating bilayer photothermal foam using a sustainable bacterial nanocellulose (BNC) and active black titania nanoparticles, and the hydrophilic three-dimensional interconnected porous network of BNC contributed to the fast evaporation of water (1.26 kg·m^−2^·h^−1^ under real-time conditions) with reduced vaporization enthalpy via the BNC–water interaction. Li et al. [33] prepared an anionic polyelectrolyte-based hydrogel (APH) as an “all-in-one” evaporator with poly(vinyl alcohol) (PVA) as the matrix and poly(3,4-ethylenedioxythiophene)/poly(sodium *p*-styrenesulfonate) (PEDOT/PSS) as the solar absorber by the freeze–thawing method. Deriving from the interfacial effect from SO_3_^−^ of PSS and OH of PVA, it was reported that the evaporation enthalpy of water from APH in water immersion was as low as 1624.14 kJ·kg^−1^ so that a high evaporation rate (2.5 kg·m^−2^·h^−1^) under one-sun radiation was exhibited. The evaporation enthalpy was determined by differential scanning calorimetry (DSC).

Cryogel is a porous material with interconnected macropores and high porosity synthesized by the polymerization of monomers or the crosslinking of polymer precursors at a temperature lower than the freezing point of the reaction mixture [29,34,35,36]. The cryogel obtained by cryo-polymerization of hydrophilic monomers has a very high swelling degree and swelling rate, which are characteristics of a high specific surface area brought by the three-dimensional penetrating hydrophilic macropores [37]. Thus, combining photothermal materials and cryogels into photothermal evaporation devices becomes a feasible strategy [38]. In our previous work [29,39,40,41], various mushroom-like devices with different hydrophilic polymers and photothermal materials were used in photothermal evaporation, and the evaporation rate varied from 1.63 to 2.20 kg·m^−2^·h^−1^ with high light-to-evaporation conversion efficiency, which is in doubt because the real evaporation enthalpy of water from those cryogels has not been actually measured and adopted for such calculations.

In this work, polyacrylamide (PAM) cryogels were sulfonated to different degrees to act as the matrix for photothermal evaporation, and in situ polymerization was performed to load polypyrrole (PPy) as the photothermal conversion material, resulting in the main part of the solar evaporation device. The umbrella-shaped commercial melamine foam with excellent water transfer and heat insulation was kept at the bottom to assemble the whole photothermal evaporation device. Compared with bulk water, the enthalpy of evaporation of water in the sulfonated cryogels was determined and greatly reduced to 1187 J g^−1^ due to the presence of sulfonate groups, which reduced the energy required for water evaporation. As a result, the sulfonated PAM cryogel obtained a photothermal evaporation rate of up to 2.50 kg·m^−2^·h^−1^ for pure water and 2.09 kg·m^−2^·h^−1^ for 14 wt% brine with excellent salt resistance performance.

## 2. Materials and Methods

### 2.1. Materials

Pyrrole (Py) was purchased from Aladdin (Shanghai, China) as an analytical reagent (AR) and used after distillation. Acrylamide (AAm, Energy Chemical, Shanghai, China), *N*, *N’*-methylenebisacrylamide (MBA, AR, Aladin, Shanghai, China), 2-hydroxy-2-methyl-4’-hydroxyethoxy-propiophenone (HMHEP, AR, Aladdin), Na_2_SO_3_ (AR, Sinopharm Chemical, Shanghai, China), FeCl_3_ (AR, Sinopharm Chemical), formaldehyde (37 wt% in H_2_O, Sinopharm Chemical), KOH (AR, Sinopharm Chemical), and calcium chloride anhydrous (AR, Xilong Scientific, Shantou, China) were used as received. Deionized water was used in all the experiments.

### 2.2. Synthesis of Sulfonated PAM Cryogels

Polyacrylamide cryogel was prepared by UV light initiating polymerization under freezing conditions. In this study, AAm (1.11 g), MBAm (0.11 g), and HMHEP (0.02 g) were dissolved together in deionized water (10 mL) and dispersed homogeneously by sonication. Then, the obtained solution was transferred to a Petri dish with a diameter of 6 cm and frozen at −15 °C for 3 h. The frozen mixture was placed in a low-temperature test chamber (DR402, Jianheng Instrument, Shanghai, China) at −10 °C, and two UV lamps (LED-0016, 30 W, 365 nm) were placed 10 cm above and below the Petri dish. Then, the polymerization occurred for 50 min by UV light initiation. After that, the obtained cryogel was taken out and soaked in deionized water for 48 h, with the deionized water exchanged every 4 h to remove the unreacted monomer and uncrosslinked polymer. Finally, the PAM cryogel was lyophilized at −50 °C for 48 h and the monomer conversion was determined according to its remaining mass.

After obtaining the dried cryogel, the PAM cryogel was sulfonated to different degrees by controlling the proportion of reactants. A dried sheet of PAM cryogel (1.1 g) was kept in 0.01 M KOH solution (200 mL) containing Na_2_SO_3_ and formaldehyde at 60 °C for 5 h to achieve a sulfonation reaction. The molar amount of Na_2_SO_3_ and formaldehyde was equal and varied at 10–50 mol% of acrylamide unit. Afterward, the sulfonated PAM cryogel was immersed in deionized water for 24 h, the deionized water being exchanged every 4 h until the solution was neutral, and then it was lyophilized at −50 °C for 48 h. According to the molar amount of Na_2_SO_3_ or formaldehyde, the sulfonated PAM cryogels were named PAM-S(0.1), PAM-S(0.2), PAM-S(0.3), or PAM-S(0.5), where PAM-S(0.1) refers to the cryogels with a molar ratio of sodium sulfite to acrylamide unit of 0.1. The molar percent of the sulfur element of the dry cryogels was determined with element analysis to evaluate the sulfonation degree.

### 2.3. Synthesis of Sulfonated PAM Composite Cryogels with Photothermal Material

The sulfonated PAM composite cryogels were prepared by in situ polymerization of Py inside PAM cryogels. A dried sheet of sulfonated PAM cryogel was soaked in the mixed solution of Py (0.25 g) and HCl solution (1 M, 145 mL) at 0 °C for 1.5 h to absorb pyrrole monomer. Then, HCl solution (1 M, 5 mL) containing FeCl_3_ (1.5 g) was added under magnetic stirring to induce oxidative coupling polymerization in an ice bath for 3 h. After being soaked in a water/ethanol mixture (50:50 in volume) and washed with deionized water, the sulfonated PAM composite cryogels (PAM-S@PPy) were finally freeze-dried. Based on the weight of dry PAM-S@PPy cryogels, the pyrrole monomer was almost converted into PPy.

### 2.4. Determination of Equivalent Evaporation Enthalpy of Water in PAM Cryogels

To accurately determine the evaporation enthalpy of water during photothermal evaporation with different PAM cryogels, a two-layer iron container was designed (Appendix A). Pure water or different swollen cryogel samples were placed on the upper layer and anhydrous calcium chloride was placed on the container button for humidity control. Then, the whole container was completely immersed in a thermostatic bath (BY-28, Donghua Kaili Technology, Shanghai, China) at a certain temperature. Bulk water or one PAM cryogel that reached swelling equilibrium were separately placed in a Petri dish with a diameter of 3 cm. To keep the same evaporation area, the side and bottom of one cryogel sheet cut to 3 cm in diameter were wrapped with tin foil, and the actual upper surface area (*S*) was checked with a camera and ImageJ software. The evaporation experiments for bulk water and swollen cryogel were conducted in dark conditions at the same temperature and humidity. Assuming that the evaporation energy input (*U*_in_) to water under these conditions was the same, the equivalent vaporization enthalpy (*E*_equ_) of water from cryogel was calculated according to Equation (1).
(1)$$U_{in}=E_{w}\Delta m_{w}/S_{w}=E_{\mathrm{equ}}\Delta m_{p}/S_{p}$$
where *E*_w_ is the vaporization enthalpy of bulk water; Δ*m*_w_ and Δ*m*_p_ are the mass change of bulk water and swollen cryogel, respectively; and *S*_w_ and *S*_p_ are the upper surface area of bulk water and swollen cryogel, respectively. All the surface areas were determined using camera recordings with the aid of ImageJ software.

### 2.5. Photothermal Evaporation with Different Sulfonated PAM Cryogels

The apparatus for photothermal evaporation is shown in Figure 1. The photothermal evaporation experiment was performed at room temperature (25–30 °C) and 35–60% humidity using a solar simulator with an AM 1.5G filter (CEL-HXF300/CEL-HXUV 300), and the light source power was kept at 1 kW⋅m^−2^ (one-sun irradiation). In the experiment, a commercial melamine sponge was shaped into an umbrella-like bottom and embedded into a PST foam, and the composite cryogel sheet (cut into 3.0 cm diameter) was kept in contact with the top of the umbrella-like melamine sponge. The whole system was then placed in a beaker with water in contact with the umbrella-like sponge bottom. The temperature of the evaporator was recorded by an infrared thermal camera (Fluke TIS40) and the mass loss during evaporation was measured by a balance (FA2204, 0.1 mg in accuracy) connected to a computer to record the mass data.

### 2.6. Characterization

The chemical structure of the cryogel was analyzed using a Fourier transform infrared spectrometer (FTIR, NICOLET-6700, Thermo Scientific, Waltham, MA, USA) with a test wavenumber range of 4000–500 cm^−1^, scan step length of 4 cm^−1^, and 16 scanning times. The cryogel morphology was analyzed by a ZEISS EVO18 tungsten hairpin filament scanning electron microscope (SEM), and the cryogel sample was freeze-dried and gold-sprayed prior to the test. The thermal stability of the cryogel was determined by thermogravimetric analysis (TGA, Q5000IR, New Castle, PA, USA) from 25 to 800 °C at a heating rate of 10 °C⋅min^−1^. The content of the sulfur element in the cryogel was determined by an elemental analyzer (VarioELIII, Frankfurt, Germany). The solar absorption efficiency (*A*) of the composite cryogel was obtained by measuring and calculating the solar reflectance and transmittance from 250 to 2500 nm using a UV-vis-NIR spectrophotometer (SOLID3700, Palo Alto, CA, USA) equipped with an integrating sphere. Then, the absorption efficiency (*A*) was calculated by *A* = (1 − *T* − *R*), where *T* and *R* are transmission and reflectivity efficiency, respectively. The contact angle between the cryogel surface and water was measured by a contact angle measuring device (Dataphysics OCA15EC, Filderstadt, Germany). Its high-speed camera was set at 100 frames per second.

The swelling behavior of the cryogel was determined from the mass of freeze-dried cryogel after swelling in deionized water. The experimental process was as follows: we freeze-dried a cryogel sheet, recorded its initial weight, soaked it in deionized water for different lengths of time, recorded the weight of the swollen cryogel, and repeated the above procedures until the mass did not change to reach swelling equilibrium. The swelling ratio (*SR*) was obtained according to Equation (2).
(2)$$SR=\frac{m_{s}-m_{0}}{m_{0}}$$
where *m_s_* and *m*_0_ are the mass of the swollen and dried cryogel, respectively.

## 3. Results and Discussion

### 3.1. Preparation of Different Sulfonated PAM Cryogels

Polyacrylamide cryogel was prepared with MBAm as a crosslinking monomer under UV light initiation at −10 °C. According to the mass of the cryogel, the monomer conversion was nearly complete and the gel fraction was above 90%. Then, the sulfonic group was introduced into the polymer chain by sulfonation reaction at 60 °C for 5 h under alkaline conditions. The whole preparation procedure with the conversion of the AM unit into a sulfonated AM unit is shown in Figure 2.

Figure 3a shows the local FTIR spectra of PAM cryogels with different sulfonation degrees. Among them, the characteristic signals of polyacrylamide are obvious, such as the C=O stretching vibration peak at 1664 cm^−1^, the N-H bending vibration signal at 1617 cm^−1^, and the methylene deformation vibration signal at 1454 cm^−1^. Due to the sulfonation, the absorbance signals changed significantly at 1538 cm^−1^ and 1045 cm^−1^ with the sulfonation degree. As the sulfonation degree of PAM cryogels increased, the N-H bending vibration signal of the secondary amide at 1538 cm^−1^ gradually became stronger and blue-shifted, indicating that the primary amide group of PAM was replaced with the secondary amide bonded to the methylene sulfonate group, leading to higher hydrophilicity. The characteristic absorbance peak at 1045 cm^−1^, assigned to S-O stretching vibration, increased as the sulfonation degree increased.

The content of the sulfur element in sulfonated PAM cryogels was determined by elemental analysis. It can be seen from Appendix A that in the sulfonation process of PAM cryogels, the ratio of formaldehyde and Na_2_SO_3_ was increased, and the sulfur element in the obtained PAM content increased. Among them, the mass fraction of the sulfur element in PAM-S(0.5) was about 7.64 wt%, and the molar percent of sulfonate-containing units was about 15.52 mol%. The results indicate that the sulfonation degree of polyacrylamide could be changed by controlling the amount of formaldehyde and Na_2_SO_3_ added.

In addition, swelling ratio (*SR*) and dynamic contact angle experiments showed that the hydrophilicity of PAM cryogels changed significantly after sulfonation. The results of the swelling experiment are shown in Figure 3b. PAM cryogels quickly absorbed water within 5 s, and their mass hardly changed after 10 s, which indicated that they had reached swelling equilibrium. Additionally, because the hydrophilic sulfonate group promotes the interaction between water and the cryogel, the equilibrium swelling ratio also increased significantly as the sulfonation degree of PAM cryogels increased. For example, *SR* of PAM-S(0.5), the sample with the highest sulfonation degree, was about 17 g⋅g^−1^, which is significantly improved compared with 11 g⋅g^−1^ for pure PAM cryogel. PAM cryogel with different sulfonation degrees was freeze-dried to carry out the drip experiment for the contact angle, and the change in the water drop contacting the cryogel surface was photographed by a high-speed camera, so as to observe the dynamic contact angle of PAM cryogel. As shown in Figure 3c, the contact angle of pure PAM cryogel is about 24° at 0.01 s after the un-sulfonated PAM cryogel contacted the water droplet, while the water droplet completely penetrated into the sulfonated PAM cryogel within 0.01 s so that contact angle could not be detected, as shown in Figure 3d.

However, observing the pore morphology of cryogels by SEM showed that the pore structure of sulfonated cryogels did not change significantly, and the representative PAM-S(0) and PAM-S(0.5) are shown in Figure 3e,f. Most of their pores are interconnected macropores (the pore diameter is mostly larger than 10 μm). With the increase in the sulfonation degree of the PAM cryogel, the pore morphology did not change obviously, but the pore diameter increased slightly (Figure 3g,h), which may be related to its swelling behavior since higher hydrophilicity might swell and soften the cryogel wall. The larger pores of cryogels with a higher swelling ratio (*SR*) were preserved after freeze-drying.

### 3.2. Preparation of Sulfonated PAM Composite Cryogels Loaded with Polypyrrole

By in situ polymerization of pyrrole adsorbed in PAM cryogels prompted with FeCl_3_ in 1 M HCl solution, polypyrrole-loaded sulfonated PAM composite cryogels were obtained and named PAM-S(*)@PPy. Since pyrrole was nearly polymerized into PPy, the amount of PPy in the composite cryogels remained the same. Shown in Appendix A is the comparison of the FTIR spectrum between PAM-S(0.2) and PAM-S(0.2)@PPy after loading polypyrrole. The appearance of FTIR signals at 1194 cm^−1^ and 928 cm^−1^, ascribed to the C-N stretching vibration and the C-H bending vibration of PPy, respectively, confirms the introduction of PPy inside PAM cryogels. After loading polypyrrole, the hydrophilicity of the PAM cryogel changed, which was related to the hydrophobicity of polypyrrole itself.

As shown in Figure 4a, the equilibrium *SR* of PAM-S@PPy cryogels was greatly reduced compared with PAM-S cryogels. However, the increase in the sulfonation degree no longer increased *SR*, contrary to the case of PAM-S cryogels. Among PAM-S@PPy cryogels, *SR* of PAM-S(0.5)@PPy is the lowest (about 7 g⋅g^−1^), while that of others is about 8 g⋅g^−1^, slightly higher than that of PAM-S(0)@PPy cryogel. This result may be due to the introduction of hydrophobic PPy inside PAM cryogel. The hydrogen bonding interaction between the sulfonate group and the pyrrole monomer may help the adsorption of pyrrole and the attachment of the produced polypyrrole to the wall surface of the PAM-S cryogel, since the sulfonate groups have an ionic interaction with protonated pyrrole and polypyrrole. Therefore, when the sulfonation degree of PAM cryogel increased, the cryogel surface covered with PPy also increased, resulting in a decrease in both hydrophilicity and *SR*. As shown in Figure 4b–f, the dynamic contact angle of PAM-S@PPy cryogel increased from 29° to 128.5° as the sulfonation degree increased. The detailed dynamic contact angle values are listed in Appendix A. At the same time, the morphology of PAM-S@PPy cryogel was observed by SEM after freeze-drying. From SEM results (Figure 4g,h), it can be found that PPy particles appeared on the wall surface of PAM-S(0.2)@PPy cryogel after loading polypyrrole. The SEM images of other PAM-S@PPy cryogels are shown in Appendix A, in which the surface of PAM-S(0.5)@PPy cryogel is rougher and more wrinkled, which means that polypyrrole covered more surface of PAM-S cryogels with a higher sulfonation degree.

Thermogravimetric analysis (TGA) is generally used to test the thermal stability of substances. As shown in Appendix A, as the sulfonation degree increased, the first decomposition temperature of PAM-S cryogels decreased from 220.2 °C to 145.8 °C due to the presence of an imide group with the bonded methylenesulfonate. After PAM-S(0.2) was loaded with polypyrrole, the first decomposition temperature decreased from 159.3 °C to 141.1 °C, caused by the loss of HCl dopant. Otherwise, the thermal degradation temperature of all PAM cryogels was higher than the temperature of photothermal evaporation.

Pyrrole-loaded PAM cryogels should absorb sunlight in a broad spectral range with little light transmittance and reflectance. Therefore, as shown in Figure 5, the sunlight absorbance spectrum was evaluated using UV-vis-NIR reflectance and transmittance spectroscopy.

As shown in Figure 5a,b, the reflectance and transmittance of PAM-S(0.2) cryogel without loading PPy in the wavelength range of 300–1500 nm are as high as 80% and 17%, respectively; thus, its absorbance (Figure 5c) is just several percentage points within this range. After loading PPy into PAM cryogels, the reflectance and transmittance of PAM-S@PPy dropped sharply. For example, the reflectance decreased to about 5% (Figure 5a), and the transmittance of the PAM-S(0.2)@PPy cryogel was close to zero (Figure 5b). The absorbance of PAM-S(0.2) cryogel before loading pyrrole is usually lower than 20% in the wavelength range of 500–1300 nm. However, after PPy was loaded into PAM cryogels, the absorbance of PAM-S@PPy became as high as 98% due to the sharp decrease in transmittance and reflectance, as shown in Figure 5C. In addition, the absorbance of different PAM-S cryogels loaded with pyrrole was greater than 97%, which reduced the interference of the factor of light absorption with its photothermal evaporation efficiency. Based on the intensity distribution of sunlight on the wavelength in Figure 5c, the total solar light absorption efficiency (*η*_abs_) of different cryogels in the range of 200–2000 nm was calculated according to Equation (3), and the results are shown in Appendix A.
(3)$$\eta_{\mathrm{abs}}=\overset{2500}{\underset{200}{\sum }}\eta_{\lambda }I_{\lambda }/\overset{2500}{\underset{200}{\sum }}I_{\lambda }$$
where *η*_λ_ and *I*_λ_ are the absorbance intensity at each wavelength (*λ*).

### 3.3. Determination of Equivalent Vaporization Enthalpy of Water in PAM Cryogels

In the process of water evaporation, water molecules escape from bulk water. In addition to overcoming the interaction force between water molecules, it is also necessary to overcome the interaction force between water molecules and the evaporation surface. Thus, water exists in three states, namely bound water (BW), intermediate water (IW), and free water (FW) [31,33,42].

If the interaction force between BW and the evaporation surface is strong, water still cannot be evaporated naturally. The force among FW molecules is strong, also causing difficulty in evaporating from bulk water [32,43]. However, IW is mainly in the small clusters of water molecules, so less energy is required for IW to evaporate from bulk water. Therefore, increasing the IW content is beneficial to reduce the energy required for water evaporation, and thereby reduce the evaporation enthalpy of water. Thus, sulfonate groups with different amounts were introduced into PAM cryogels to increase their hydrophilicity and evaluate their capacity in the modulation of the water state based on the real evaporation enthalpy of water from the cryogels.

The evaporation enthalpy of water has been determined with differential scanning calorimetry (DSC) in the photothermal evaporation literature [29,33,44] for the appraisement of light-to-evaporation conversion efficiency. However, the obtained value of evaporation enthalpy from DSC was measured in a wide range of temperatures and did not reflect the real energy needed for water evaporation at a definite temperature. Thus, the self-made apparatus and the designed procedure were used to measure the real evaporation enthalpy of water from cryogels. The temperature in the thermostatic bath and inside the container were 42.2 °C and 41.8 °C, respectively. Based on the mass loss of bulk water and PAM cryogels at the same temperature and humidity in the dark (Figure 6a,b), equivalent vaporization enthalpy (*E*_equ_) of water from PAM cryogels was obtained.

As shown in Figure 6a, with bulk water as the control, the weight loss from PAM-S(x) cryogels is obviously different and dependent on the sulfonation degree. The weight loss at each instant increases gradually with the sulfonation degree, with PAM-S(0.5) cryogel being the highest. After 2 h, the mass loss of all samples reached a steady state. Thus, linear fitting of mass loss–time data was performed starting at 2 h and ending at 6 h, as shown by the dotted line in Figure 6a, whose fitting minimum variance of R^2^ confirms the reliability of the measured *E*_equ_. Based on the slope of the fitting line, the equivalent evaporation enthalpy (*E*_equ_) of water from PAM-S(x) cryogel at 42 °C was obtained while the evaporation enthalpy of bulk water at the same temperature was taken as 2403 J⋅g^−1^ [45]. The results from different PAM-S(x) cryogels are shown in Figure 6c. It can be seen from the data that the equivalent evaporation enthalpy decreases gradually with the increase in the sulfonation degree. *E*_equ_ of PAM-S(0.5) cryogel is only 1187 J⋅g^−1^, being below half of evaporation enthalpy of pure water. This result suggests that the hydrophilic sulfonate group promotes the activation of water in the cryogel and increases the content of intermediate water (IW). The experiments were repeated several times to check the reproducibility of the results. Although the temperature and humidity could not be kept exactly the same, the results kept the same tendency.

The equivalent evaporation enthalpy (*E*_equ_) of water from PAM-S@PPy cryogels was also determined in the same way. The mass loss with time, the linear fitting line from 2 h to 6 h, and the values of *E*_equ_ are shown in Figure 6a–c, respectively. Due to the hydrophobic polypyrrole, the equivalent evaporation enthalpy of water in PAM cryogel does not monotonically decrease with the increase in the sulfonation degree, with PAM-S(0.2)@PPy having the lowest value of 1304 J⋅g^−1^. This phenomenon might suggest that the hydrophilicity/hydrophobicity of the matrix should be optimal to reach the lowest evaporation enthalpy of water. High hydrophilicity would lead to a high percentage of bound water while high hydrophobicity would lead to a high percentage of free water, both of which adversely affect the decrease in evaporation enthalpy. With the lowest equivalent evaporation enthalpy, PAM-S(0.2)@PPy cryogel was expected to have the best performance in subsequent photothermal evaporation.

### 3.4. Photothermal Evaporation of Water from Different Cryogels and Light-to-Evaporation Conversion

To improve the utilization efficiency of light energy and increase the photothermal evaporation rate in the photothermal evaporation process, the photothermal evaporation device was designed as shown in Figure 1. Commercial melamine foam was cut into an umbrella-like shape and embedded in polystyrene foam, with PAM cryogel discs placed on top of the melamine foam. The thermal conductivity coefficients of melamine foam and polystyrene foam were 0.043 and 0.0367 W⋅m^−1^⋅K^−1^, respectively, much lower than that of pure water (0.592 W⋅m^−1^⋅K^−1^). Low thermal conductivity reduced the transfer of thermal energy to the water body below in the beaker and PST foam avoided possible light exposure to the water body. Due to the excellent swelling ratio and swelling rate of melamine foam and PAM cryogels, the tip of the melamine foam could transfer water to the upper surface of the cryogel quickly after contacting the water, and the lack of water supply during photothermal evaporation was avoided.

The mass loss of the photothermal evaporation system under simulated one-sun irradiation (incident power = 1 kW⋅m^−2^) was recorded, and Figure 7a shows one example of the mass change from each cryogel in the first hour. It can be seen that the mass loss rate of water gradually accelerates in the first ten minutes, and finally tends to be stable. This is because the light energy is needed to heat the photothermal evaporation system in the beginning. When the surface temperature of the cryogel reaches a steady state, the mass loss of the photothermal evaporation system tends to increase linearly with time. The long-term performance of cryogels is crucial for photothermal evaporation performance, and Figure 7b shows the mass loss during the first 5 h under one-sun light. The evaporation rate was calculated every half hour, and Figure 7c shows that the evaporation rate does not decrease significantly over time, indicating that the device is capable of continuous photothermal evaporation of water.

The evaporation rates (*R*_evp_) of water from different PAM cryogels are listed in Table 1. The evaporation rate of PAM-S(0.2)@PPy cryogel reached the highest value of 2.50 kg⋅m^−2^⋅h^−1^ among the tested cryogels, while un-sulfonated PAM-S(0)@PPy cryogel had a rate of 1.58 kg⋅m^−2^⋅h^−1^. This shows that the introduction of sulfonate groups could increase the evaporation rate of PAM cryogels. In addition, the evaporation rate has the same increasing sequence as the equivalent evaporation enthalpy, indicating the importance of optimal hydrophilicity of the matrix for photothermal evaporation. In addition, the evaporation rate of PAM-S(0.2) cryogel without polypyrrole is only 0.83 kg⋅m^−2^⋅h^−1^, indicating that polypyrrole provides important support for the conversion of solar radiation into heat energy. Light-to-heat conversion increases the temperature of the cryogels’ surface during photothermal evaporation to a steady value (*T*_steady_), as shown in Figure 7d–h. Furthermore, the initial temperature of the cryogel surface before photothermal evaporation (*T*_initial_) was recorded for each cryogel test, as shown in Appendix A, considering the variation in environmental temperature throughout the day. All the temperature values are summarized in Appendix A. *T*_steady_ for different cryogels varies from 41.7 to 44.2 °C, with that of PAM-S(0.2)@PPY being 42.4 °C. High temperature causes low evaporation enthalpy, leading to a high photothermal evaporation rate.

When the cryogels are in a stage of photothermal evaporation, the energy absorbed from solar radiation (*I*_abs_) is utilized in three aspects: the heating of water (*Q*_heating_) from room temperature to that of the cryogel surface, the evaporation *(Q*_evap_) of water, and the heat exchange (*Q*_loss_) between the evaporation device and the environment. In this way, according to the principle of energy conservation, Equation (4) is obtained.
(4)$$\eta_{\mathrm{abs}}I_{\mathrm{inc}}=I_{\mathrm{abs}}=\left(R_{\mathrm{evap}}-R_{\mathrm{dark}}\right)\times \left(E_{equ}+C_{w}\times \Delta T\right)+Q_{\mathrm{loss}}$$

*R*_evap_ is the evaporation rate under light irradiation while *R*_dark_ is that of pure water in the dark (determined to be 0.350 kg⋅m^−2^⋅h^−1^), and their difference (*R*_light − _*R*_dark_) is the actual photothermal evaporation rate (*R*_real_) of cryogels. *I*_inc_ is the intensity of incident light (1 kW⋅m^−2^), and *η*_abs_ is the light absorption efficiency. *C*_w_ is the specific heat capacity of water, being 4.18 J⋅g^−1^⋅K^−1^. Δ*T* is the temperature rise of water during photothermal evaporation, equal to (*T*_steady − _*T*_initial_). *E*_equ_ is the equivalent enthalpy of evaporation of water from cryogels, with the data shown in Figure 6c (neglecting the change in enthalpy of evaporation of water under a small temperature difference). In the experiment, the heat insulation design greatly reduces the heat loss to the environment, so the heat loss (*Q*_loss_) in Equation (4) is negligible. Therefore, evaporation efficiency (*η*_evap_) and total efficiency (*η*_total_) can be calculated according to Equations (5) and (6), respectively. The results are shown in Table 1.
(5)$$\eta_{\mathrm{evap}}=R_{\mathrm{real}}\times \frac{\left(E_{\mathrm{equ}}+C_{w}\times \Delta T\right)}{I_{\mathrm{abs}}}$$
(6)$$\eta_{\mathrm{total}}=R_{\mathrm{real}}\times \frac{\left(E_{\mathrm{equ}}+C_{w}\times \Delta T\right)}{I_{\mathrm{inc}}}$$

The equivalent enthalpy of evaporation (*E*_equ_) is used as the actual evaporation enthalpy of water during photothermal evaporation. Although the evaporation processes were not exactly at the same temperature, the temperature difference is very small, so the difference in evaporation enthalpy is negligible. PAM-S(0.2)@PPy cryogel not only has the fastest evaporation rate but also the lowest *E*_equ_, resulting in the highest total evaporation efficiency of 80.8% (Table 1). Although PAM-S(0.5)@PPy has the second highest evaporation rate, its *E*_equ_ is also rather low, being the second lowest, leading to the lowest evaporation efficiency of 59.8%.

Compared with previous reports, the values of evaporation efficiency obtained in this work are much lower. It is reasonable that those values of evaporation enthalpy used to evaluate evaporation efficiency in other reports were those of pure water or those from DSC measurement and are much larger than our values. Moreover, PAM-S(0.5)@PPy cryogel has the second lowest value of *E_equ_* (1398 J⋅g^−1^), but its higher hydrophobicity based on the contact angle to water and lower water transport ability based on swelling behavior cause much lower evaporation rate and efficiency. Strengthening sunlight absorbance, accelerating water uptake and transport, and decreasing water evaporation heat are three keys to photothermal evaporation. However, by summarizing the literature and checking our findings (Table 1), it should be emphasized that decreasing water evaporation heat is the most dominant, since very high sunlight absorbance, water uptake, and water transport have been achieved by different groups.

### 3.5. Simulated Seawater Desalination and Condensate Collection

PAM-S(0.2)@PPy cryogel with the best water evaporation performance was used to conduct further experiments to explore its practical application in solar-driven seawater desalination with NaCl solution as artificial seawater. Figure 8a,b shows the water loss and evaporation rate changes for 3.5 and 14.0 wt% NaCl solutions, respectively, with the present evaporation device within 5 h.

As for 3.5 wt% NaCl solution, the evaporation rate was as high as 2.49 kg⋅m^−2^⋅h^−1^ under one-sun radiation. Further increasing the brine concentration to 14 wt%, the evaporation rate decreased to 2.09 kg⋅m^−^^2^⋅h^−^^1^. There was no apparent salt crystallization on the cryogel surface after three days of repeated 5-hour evaporation of the high-concentration saline (Appendix A), which was caused by the hydrophilic cryogel to effectively prompt the transport of saline solution and avoid salting-out on the cryogel. The condensate from seawater evaporation was collected using a custom-designed water harvesting device (Figure 8c), and the quality of condensed water was assessed by inductively coupled plasma mass spectrometry (ICP-MS). After desalination by our solar water evaporation system, the concentration of Na^+^ was significantly reduced and lower than that of normal drinking water (Figure 8d).

## 4. Conclusions

In summary, PAM cryogels with interconnected macropores were prepared by UV-light-induced low-temperature polymerization and sulfonated in KOH solution containing formaldehyde and Na_2_SO_3_. FTIR and elemental analysis confirmed the successful introduction of sulfonate groups onto the PAM cryogels and the modulation of sulfonation degree by the feed ratio of formaldehyde to Na_2_SO_3_ to change the hydrophilicity. Polypyrrole, as a photothermal conversion material, was loaded into sulfonated PAM cryogels to enhance high solar light absorption but induced a decrease in hydrophilicity depending on the sulfonation degree based on the results of swelling behavior and contact angle of PAM-S@PPy cryogels. To evaluate the parameters affecting photothermal evaporation performance, the real evaporation enthalpy of water from PAM-S@PPy cryogels was determined in the dark under similar temperature and humidity conditions, showing that PAM-S(0.2)@PPy composite cryogel exhibited the lowest equivalent vaporization enthalpy of 1304 J⋅g^−1^. Both the sulfonation and loading behavior of PPy affected the equivalent vaporization enthalpy from PAM-S@PPy composite cryogels. The composite cryogels and mushroom-like melamine foam as the substrate constituted the evaporation device and exhibited an evaporation rate of up to 2.50 kg⋅m^−2^⋅h^−1^ within 5 h under one-sun irradiation, with the total photothermal conversion efficiency from irradiated light to evaporated water being about 80.8%, which is relatively high but reliable since the actual vaporization enthalpy of water was used. In the simulated seawater desalination, PAM-S(0.2)@PPy cryogel had evaporation rates of 2.49 kg⋅m^−2^⋅h^−1^ and 2.09 kg⋅m^−2^⋅h^−1^ for 3 and 14 wt% NaCl solution, respectively, without salting-out. Therefore, our findings based on the determination of real evaporation enthalpy disclosed the significance of lessening evaporation enthalpy in photothermal evaporation and the importance of actual vaporization enthalpy of water for the determination of photothermal evaporation efficiency, providing a valuable idea for increasing the upper limit of the photothermal evaporation rate.

## Figures and Tables

**Figure 1 polymers-15-02108-f001:**
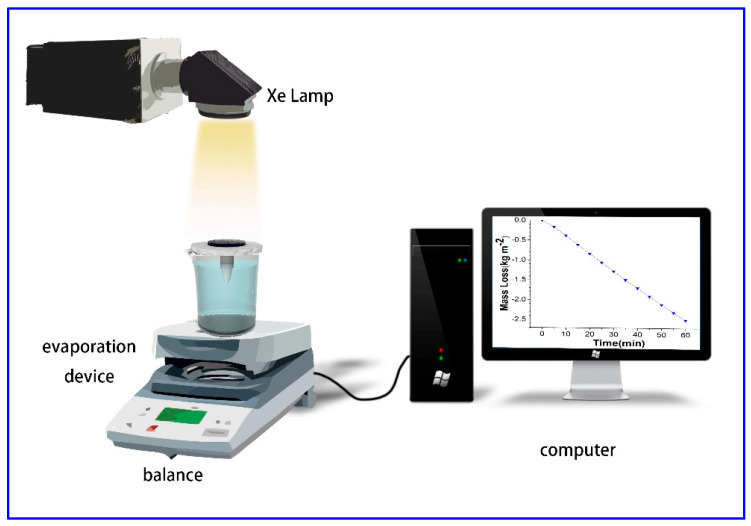
The schematic setup for the photothermal evaporation measurement.

**Figure 2 polymers-15-02108-f002:**
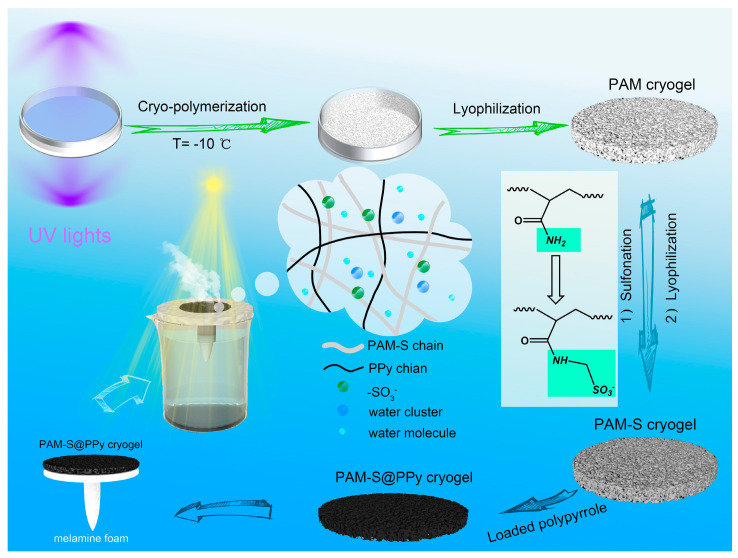
The preparation procedure of PAM-S(x)@PPy cryogels for photothermal evaporation.

**Figure 3 polymers-15-02108-f003:**
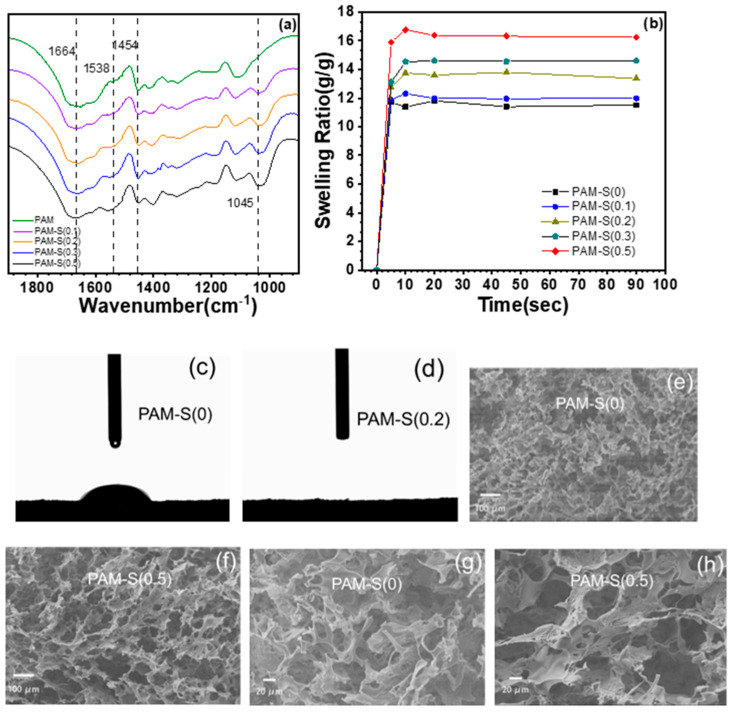
(**a**) Local FTIR spectra of PAM cryogels with different sulfonation degrees; (**b**) swelling ratio test of different sulfonated PAM cryogels; (**c**,**d**) dynamic contact angle determination of PAM-S(0) and PAM-S(0.2) cryogels at 0.01 s; (**e**–**h**) SEM images of PAM-S(0) and PAM-S(0.5) cryogels at different magnifications.

**Figure 4 polymers-15-02108-f004:**
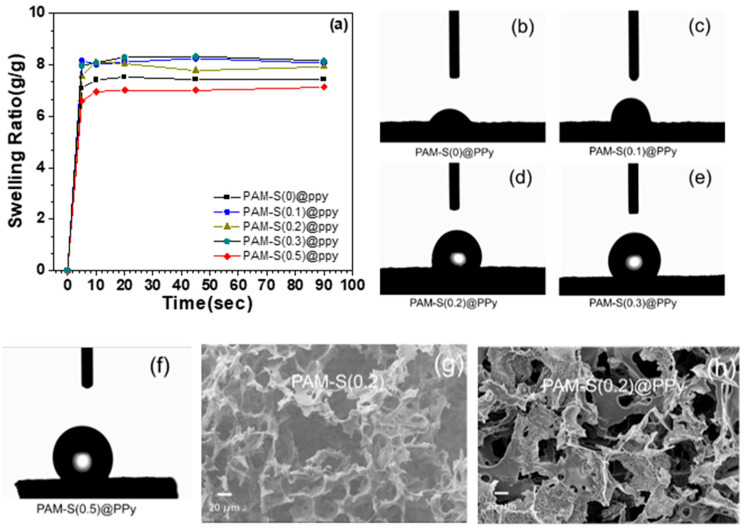
(**a**) Swelling ratio of different sulfonated PAM cryogels loaded with PPy; (**b**–**f**) dynamic contact angle determination of sulfonated PAM cryogels loaded with PPy at 0.01 s; (**g**,**h**) SEM images of PAM-S(0.2) cryogels before and after loading pyrrole.

**Figure 5 polymers-15-02108-f005:**
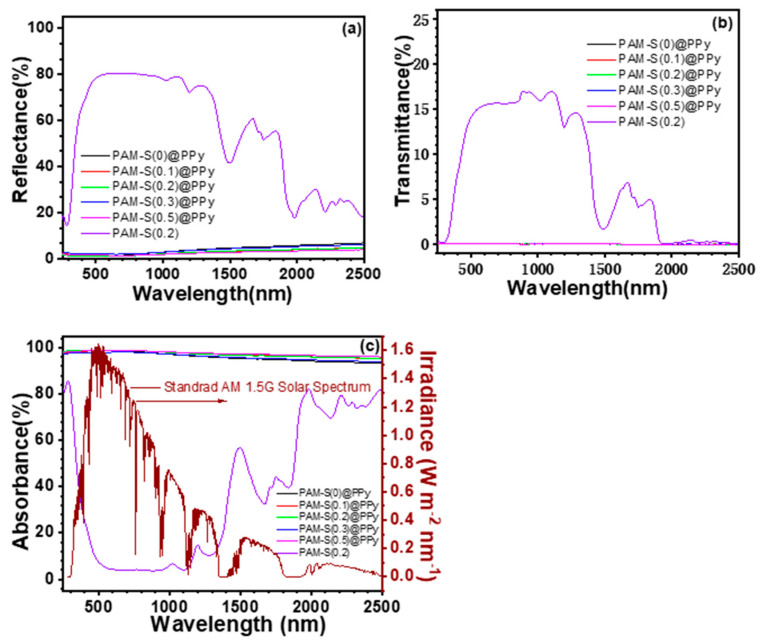
UV-vis-NIR reflectance (**a**), transmittance (**b**), and absorbance ((**c**), left Y-axis) of cryogels with solar spectral radiance ((**c**), right Y-axis).

**Figure 6 polymers-15-02108-f006:**
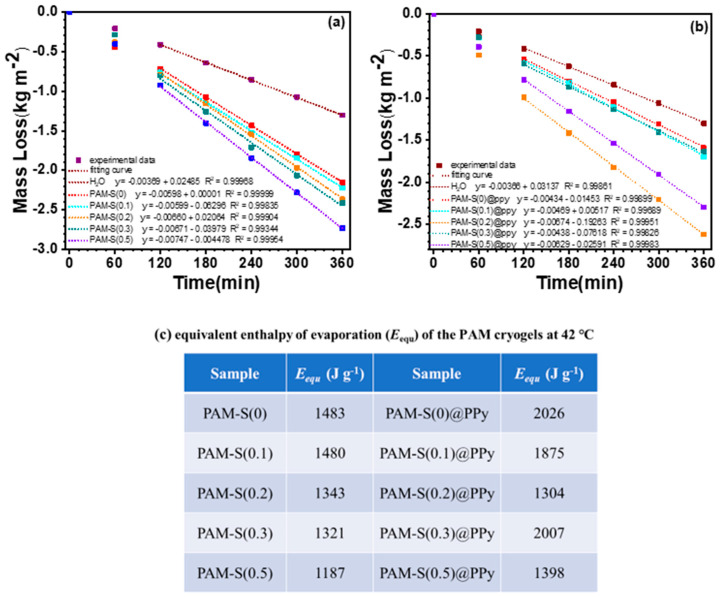
Mass loss of PAM-S cryogels without (**a**) and with (**b**) PPy at the same temperature (42 °C) and humidity in the dark (solid square: experimental data, dotted line: linear fitting result from 120 to 360 min); (**c**) equivalent enthalpy of evaporation (*E*_equ_) of PAM cryogels.

**Figure 7 polymers-15-02108-f007:**
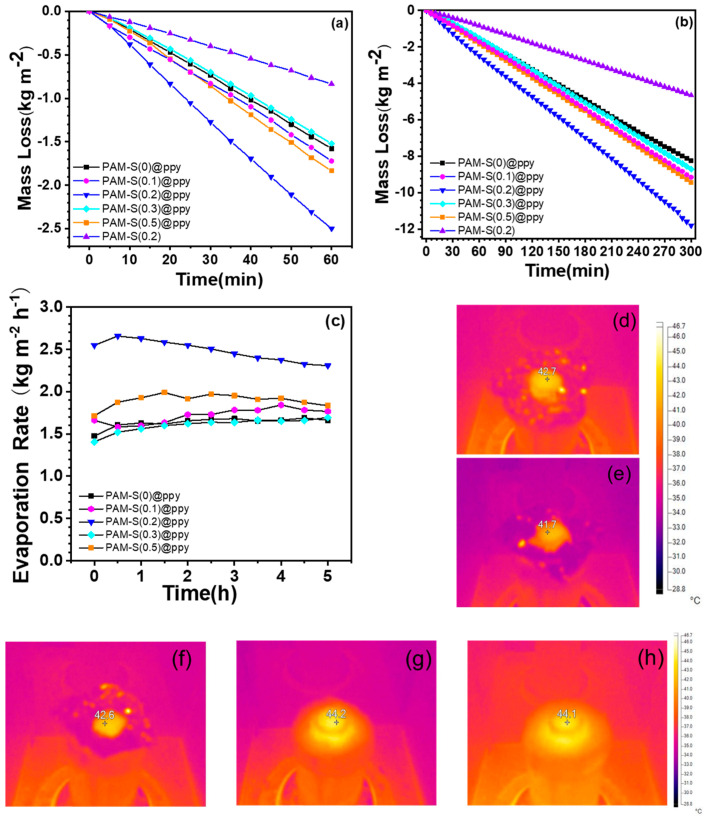
The weight loss of different PAM cryogels within the first hour (**a**) and 5 h (**b**) under one-sun irradiation; (**c**) evaporation rate of cryogels every 30 min within the first 5 h; (**d**–**h**) infrared camera pictures of PAM-S(0)@PPy, PAM-S(0.1)@PPy, PAM-S(0.2)@PPy, PAM-S(0.3)@PPy, and PAM-S(0.5)@PPy cryogel surface.

**Figure 8 polymers-15-02108-f008:**
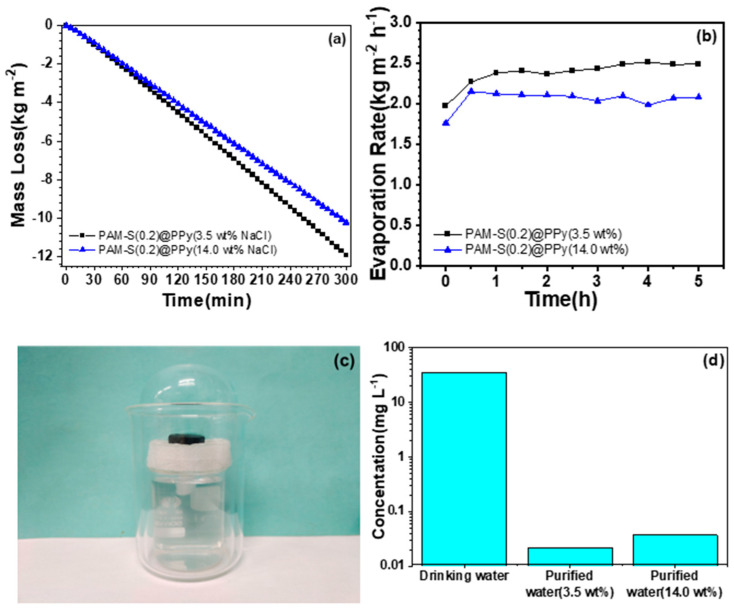
Weight loss (**a**) and evaporation rate (**b**) of PAM-S(0.2)@PPy cryogel within the first 5 h under one-sun irradiation; (**c**) water harvesting device for the simulated seawater desalination; (**d**) the concentration of Na^+^ in the collected water.

**Table 1 polymers-15-02108-t001:** Summary of photothermal evaporation parameters under one-sun irradiation.

Cryogel	*R*_evap_ (kg⋅m^−2^⋅h^−1^)	*R*_real_ (kg⋅m^−2^⋅h^−1^)	*η*_evap_ (%)	*η*_total_ (%)
PAM-S(0)@PPy	1.58	1.23	72.3	70.8
PAM-S(0.1)@PPy	1.72	1.37	74.4	73.2
PAM-S(0.2)@PPy	2.50	2.15	82.1	80.8
PAM-S(0.3)@PPy	1.52	1.17	68.8	67.1
PAM-S(0.5)@PPy	1.83	1.48	60.8	59.8

## Data Availability

Not applicable.

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
