# Peer review of "Sulfonated PAM/PPy Cryogels with Lowered Evaporation Enthalpy for Highly Efficient Photothermal Water Evaporation"

_polymers, 2023, doi:10.3390/polym15092108_

Round 1

Reviewer 1 Report

The paper deals with the evaluation of practical usefulness of synthesized macroporous polyacrylamide-polypyrrole cryogels for photothermal water evaporation and desalinization. The subject is topical and the final values of some of the prepared materials seem promising.

However, the manuscript should be improved as some points seem either unclear or contradicting. Please, see the comment below.

1. One of the main keystones is the selection of cryogel composition, and PAM-S(0.2)@PPy (in authors' notation) seem to be most advantageous among other tested compositions. However, the rates of evaporation in Fig. 6 and 7 for the prepared compositions differ significantly for PPy-loaded cryogels, while PAM-S(0.2)@PPy in Figs. 6 and 7 are the same, the efficiency of other gels is not so good in Fig. 7 as in Fig. 6. And this difference cannot be accounted for different conditions, in my opinion. And this data also do not fully agree with the efficiency tables later.

2. The same is about the dynamics of the unloaded PAM-S(0.2) - too much differences between Figures 6 and 7 compared to its loaded analogue, which is not explained.

3. I believe that the algorithm of calculation of enthalpies is not complete" how many replicate measurements were made, how the results were averaged - by the same time points or by for building calibration plots? what are the ranges for calculations if the least-square fits were not used, the plots are not linear.

4. In addition to this, reproducibility should be discussed anyway and the error bars should be added to all the experimental plots. Without this information, the reproducibility of finding the enthalpies and efficiencies is not shown.

5. Tables shows as Figure 6c shows too many significant digits, which cannot ve obtained from plots 6 and 7. The precision of measurements of mass losses and surface areas should be given and the uncertainty provided.

6. As a whole, such a tremendous difference between enthalpies is not obvious from plots and calculations and the explanation that the 'nature of PPy' results in such a nonlinear behaviour is not proven and the explanation is not satisfactory.

7. Significant digits should be double-checked and in most cases shortened or explained. Throughout the text.

8. Conclusions section is either the continuation or repeated information of the Discussion. In my opinion, the repetition should be removed (it can be used in Abstract) and outlooks and strong a weak points of the approach in general should be added.

9. Figure 5c is not used and it is in fact the repetition of Figure 5b. If this is absorption, it cannot be in %, in this case this is normalized absorbance using a reference point.

10. IR spectra (Fig. 3) are not very informative. The whole spectra in this form can be moved to the Supplementary, and the figure 3 should be a more appropriate absorbance (not transmittance) spectra for the range 1700-900 cm-1 with all the bands discussed in the text marked.

11. The English is clear but some mistakes and formatting errors should be corrected throughout the text.

Author Response

Comments and Suggestions for Authors (Reviewer 1)

The paper deals with the evaluation of practical usefulness of synthesized macroporous polyacrylamide-polypyrrole cryogels for photothermal water evaporation and desalinization. The subject is topical and the final values of some of the prepared materials seem promising. However, the manuscript should be improved as some points seem either unclear or contradicting. Please, see the comment below.

  1. One of the main keystones is the selection of cryogel composition, and PAM-S(0.2)@PPy (in authors' notation) seem to be most advantageous among other tested compositions. However, the rates of evaporation in Fig. 6 and 7 for the prepared compositions differ significantly for PPy-loaded cryogels, while PAM-S(0.2)@PPy in Figs. 6 and 7 are the same, the efficiency of other gels is not so good in Fig. 7 as in Fig. 6. And this difference cannot be accounted for different conditions, in my opinion. And this data also do not fully agree with the efficiency tables later.

Response: Thank you for pointing out this phenomenon in our study. Both Figure 6a and 6b are concerned with the determination of water evaporation enthalpy under dark condition, based on the evaporation rate of water from different cryogels. Those cryogels are divided into two kinds, one without the introduction of polypyrrole (PPy) and another with PPy. Thus, attention should be paid to their difference in hydrophilicity, where PAM-S(x) is more hydrophilic than PAM-S(x)@PPy.

Figure 6a shows the measurement results for PAM-S(x) without PPy and the evaporation enthalpy decreases with the sulfonation degree. Meanwhile, Figure 6b shows those of PAM-S(x)@PPy with PPy and the evaporation enthalpy is the lowest for PAM-S(0.2)@PPy with the highest evaporation rate. The influences of sulfonation and PPy introduction on the evaporation enthalpy are discussed in the manuscript. However, Figure 7 shows the photothermal evaporation results of different PPy-loaded PAM-S(x)@PPy cryogels, and the experiment was conducted at room temperature under one-sun irradiation, with the purpose of the determination of photothermal evaporation performance as well as “the efficiency”. The evaporation rates from Figure 6b and Figure 7 have the same order regarding to the sulfonation degree. As for “the efficiency” in Table 1 has not the same order since its calculation was based on our measured values of evaporation enthalpy. For an example, PAM-S(0.5)@PPy has the second highest evaporation rate but its evaporation enthalpy is also the second lowest, resulting “the efficiency” of PAM-S(0.5)@PPy in Table 1 is the lowest.   We hope that the above explanation would be clear enough.

  1. The same is about the dynamics of the unloaded PAM-S(0.2) - too much differences between Figures 6 and 7 compared to its loaded analogue, which is not explained.

Response: Thank you for pointing out this irregular phenomenon. Firstly, the data of PAM-S(x) in Figure 6A and PAM-S(x)@PPy in Figure 6B is compared. The evaporation rate (the slope of fitting line) of PAM-S(x) is lower than that of PAM-S(x)@PPy after the loading of more hydrophpbic PPy. As mentioned in our manuscript, the evaporation rate is determined by the sulfonation degree of PAM and the loading behavior of PPy onto PAM cryogel surface. We also noticed that PAM-S(0.2)@PPy had litter higher rate than that of PAM-S(0.2) and the decrease tendency for other analogues was also different. We repeated such experiments for evaporation enthalpy and obtained parallel results. We also wonder those observation and assume that the spatial effect in a limited space might be the caused besides of the hydrophilicity of cryogel matrix. As for Figure 7, PAM-S(0.2) without PPy can not absorb light and convert light energy to heat effectively, leading to its much lower evaporation rate compared with PAM-S(0.2)@PPy.    We hope that the above explanation would dispel the doubt.

  1. I believe that the algorithm of calculation of enthalpies is not complete "how many replicate measurements were made, how the results were averaged - by the same time points or by for building calibration plots? what are the ranges for calculations if the least-square fits were not used, the plots are not linear.

Response: We thank the reviewer for pointing out this issue. For the determination of the evaporation enthalpy, we repeated the experiment many times, and their results have very slight differences. Because the experimental conditions are not accurately the same (including temperature and humidity), so the results are not suitable for averaging. However, their relative values and trend are the same. The result of the enthalpy of evaporation is calculated according to the line slope from linear fitting as shown in Figure 6 of the revised manuscript. We have changed the solid line of data points to the dot line of linear fitting data and the slope values of fitting line have been offered. The fitting range is from 120 to 360 min. Thus, we recalculated the evaporation enthalpy according to the updated slopes. Thus, both Figure 6c and Table 1 have their changes accordingly.

  1. In addition to this, reproducibility should be discussed anyway and the error bars should be added to all the experimental plots. Without this information, the reproducibility of finding the enthalpies and efficiencies is not shown.

Response: We appreciate highly this delightful issue. As for the determination of equivalent evaporation enthalpy, we followed the long-term evaporation and recorded the data at different intervals. With the linear fitting of those data, we obtained final values of enthalpy and fitting errors have been offered in the updated Figure 6, showing the results are convincible. As mentioned the above responses, the measurement has been repeated several times. However, due to difficulty in accurate controlling of temperature and humidity, the different results is not appropriate to be averaged. As well, as for the photothermal evaporation, all the experiments were performed in open air within successive days. However, their environmental temperature and humidity were occasional. Thus, in spite of repeating photothermal evaporation experiment to check the reproducibility, the obtained results is also not appropriate to be averaged. Thus, error bars have not been given to the figures.    We hope for the understanding from the reviewers.

  1. Tables shows as Figure 6c shows too many significant digits, which cannot ve obtained from plots 6 and 7. The precision of measurements of mass losses and surface areas should be given and the uncertainty provided.

Response: Great thanks for those comments. We have checked the significant digits of the full text to ensure that the significant digits of the same type of data are consistent. In the experiment, the mass was obtained by electronic balance (accuracy is 0.1 mg), and the area is calculated by the software Imagine J from the digital photo of the cryogels. Thus, the information has added at the end of in updated Subsection 2.4.

  1. As a whole, such a tremendous difference between enthalpies is not obvious from plots and calculations and the explanation that the 'nature of PPy' results in such a nonlinear behaviour is not proven and the explanation is not satisfactory.

Response: We are joyful for this open comment. As shown in Figure 6a and 6b, the slopes of fitting lines for PAM-S(x) are much higher than that of pure water while the slope difference between pure water and PAM-S(x)@PPy except PAM-S(0.2)@PPy and PAM-S(0.5)@PPy is relatively lower. In our manuscript, we have discussed the crossover influences of sulfonation (increasing the hydrophilicity) and loading PPy (decreasing the hydrophilicity). Indeed, the hydrophilic sulfonate groups can reduce the enthalpy of evaporation, as shown in Figure 6A. The evaporation enthalpy decreases with the increase of sulfonation degree of PAM cryogels. However, the behavior of loading PPy into different PAM cryogels was affected by its sulfonation degree. For example, as demonstrated with Figure 3 and Figure 4, PAM-S(0.5) cryogel with the highest sulfonation degree has the highest hydrophilicity, but after loading hydrophobic PPy, PAM-S(0.5)@PPy was the least hydrophilic according to its swelling rate and dynamic contact angle. Therefore, we believe that the evaporation enthalpy of PAM cryogels loaded with PPy is jointly affected by the hydrophilic sulfonate group and the hydrophobic polypyrrole. Those experimental results reflects that the 'nature of PPy' results in such a nonlinear behaviour.   We hope that our response would dispel the reviewer doubt.

  1. Significant digits should be double-checked and in most cases shortened or explained. Throughout the text.

Response:  Great thanks for this applicable comment. We've checked the whole text of our manuscript and kept significant digits reasonable and consistent.

  1. Conclusions section is either the continuation or repeated information of the Discussion. In my opinion, the repetition should be removed (it can be used in Abstract) and outlooks and strong a weak points of the approach in general should be added.

Response: We are very appreciated with this useful suggestion. We have revised the Conclusions section, eliminated the repetition from Abstract section and clarified of the experimental approach and research findings.

  1. Figure 5c is not used and it is in fact the repetition of Figure 5b. If this is absorption, it cannot be in %, in this case this is normalized absorbance using a reference point.

ResponseThank you for pointing out this problem. Indeed, Figure 5C was just mentioned once in the previous manuscript. It comes from Figure 5a and 5c, as indicated by “A = (1 − TR)” in Subsection 2.6. Thus, it is not the repetition of Figure 5b. Indeed, it is referred to the light absorbance (A) and its definition “A = (1 − TR)” has been given differently from what the reviewer thought. Thus, it can not be normalized using a reference point. Additionally, the total solar light absorption efficiency (ηabs) of different cryogels in the range of 200–2000 nm was calculated according to equation (3) and based on the intensity distribution of sunlight (shown in Figure 5c). To protrude Figure 5c, we have added discussions about the absorbance of PAM-S(0.2) cryogel and mentioned Figure 5c to PAM-S@PPy cryogels. We hope that our explanation would be positive.

  1. IR spectra (Fig. 3) are not very informative. The whole spectra in this form can be moved to the Supplementary, and the figure 3 should be a more appropriate absorbance (not transmittance) spectra for the range 1700-900 cm-1 with all the bands discussed in the text marked.

ResponseThank you for pointing out this problem. We have adjusted the range of Figure 3a to show the local IR spectrum.

  1. The English is clear but some mistakes and formatting errors should be corrected throughout the text.

ResponseWe thank the reviewer for pointing out this issue. We have checked the whole article and corrected the errors.

Reviewer 2 Report

-          Explain AR acronym  - Pyrrole (Py) were purchased from Aladdin as AR reagent and used after distillation. 

-          the cryogel and sulfonated cryogel structure must be presented

-          Py (0.25 g) and HCl solution (1 M, 145 mL) at 0 °C for 1.5 h to absorb 127 pyrrole monomer. Then HCl solution (1 M, 5 mL) containing FeCl3 (1.5 g) – the ratio Py/initiator was 0.25 / 1.5?

-          The structures of the synthesised compounds including the composites must be presented.

-          Figure 2 – has errors in representation

-          the characterization of the compounds is presented without establishing a connection and interdependence between the methods

-          FTIR spectra are not commented on the entire registered domain and without assigning the peaks corresponding to the present functional groups

-          For example, SR of PAM-S(0.5), the  sample with the highest sulfonation degree, is about 17 g×g-1 , which is significantly improved compared with 11 g×g-1 for pure PAM cryogel. – why? Explain

-          un-sulfonated = non-sulfonated

-          With the increase of  sulfonation degree of PAM cryogel, the pore morphology even did not change obviously, but the pore diameter increased slightly (Figure 3g and 3h), which may be related to its  swelling behavior. The larger pores of cryogels with higher swelling ratio (SR) were preserved after freeze-drying - the pore diameter depends on sulfonation degree, crosslinking degree?

-          The interaction between sulfonate group and pyrrole monomer helps the adsorption of pyrrole and the attachment of produced polypyrrole on the surface of PAM-S cryogel – authors must present the chemical adsorption of pyrrole onto the cryogel chemical surface and how this adsorption helps the polymerisation of pyrrole

-          From SEM results (Figure 4g and h), it can be found that particles appeared on the surface of PAM-S(0.2)@PPy cryogel after loading polypyrrole. – which particles?

in Supporting Information

-          Composition comparison of sulfonated PAM before and after PPy loading – to clearly highlight the changes made in PAM-S(0.2)@PPy cryogels in fig. S2   the FTIR spectrum of PPy  must be insert

-          Figure S4. Thermogravimetric analysis results of PAM-S(0) (a), PAM-S(0.2) (b), PAM-S(0.2) (c), PAM-S(0.3) (d), PAM-S(0.5) (e) and PAM-S(0.2)@PPy (f) – what it means weight retention?

Author Response

Comments and Suggestions for Authors (Reviewer 2)

- Explain AR acronym - Pyrrole (Py) were purchased from Aladdin as AR reagent and used after distillation.

Response: Thank the reviewer for pointing out this problem. We have corrected the error, and AR is analytical reagent.

- the cryogel and sulfonated cryogel structure must be presented.

Response: The comment is very helpful. In Figure 2, the structure of AM unit for the cryogel and that of sulfonated AM for “sulfonated cryogel” have been absence. We have changed the color of those structures to highlight them. As well, the chemical structures and surface morphologies of the different cryogels are well characterized with FTIR, elemental analysis and SEM in our manuscript.

- Py (0.25 g) and HCl solution (1 M, 145 mL) at 0 °C for 1.5 h to absorb pyrrole monomer. Then HCl solution (1 M, 5 mL) containing FeCl3 (1.5 g) – the ratio Py/initiator was 0.25 / 1.5?

Response: Thank you for this comment. First of all, it should be mentioned that FeCl3 for the oxidative coupling polymerization of pyrrole is the oxidant but not the initiator. Then, the weight ratio Py to FeCl3 was indeed 0.25/1.5, suggesting the excessive amount of FeCl3 to ensure the complete conversion of Py to its polymer. Additionally, this ratio of monomer to oxidant was set with reference to the articles [1-3].

- The structures of the synthesised compounds including the composites must be presented.

Response: Thank you for this comment, being similar to previous one. The structure of AM unit for the cryogel and that of sulfonated AM for “sulfonated cryogel” have been mentioned Figure 2 to emphasize the hydrophilicity enhancement with sulfonation. As for polypyrrole, it is a well-known conjugated polymer and it is not urgently needed to offer its chemical structure. The composition of sulfonated PAM have been characterized and determined with different methods in our manuscript, while Table S1 offers the information of sulfur element weight percent in its second column. However, we found out one error in the previous Table S1 after we discussed and checked the calculation. “S (mol%)” was wrongly used but it is the molar percent of sulfonated acrylamide unit in polymer. Thus, we have modified the Table S1 and added one Subsection of “Molar percent of sulfonated AM unit in the crosslinked polyacrylamide”. In the updated Table S1, the third column of “S (mol%)” is the molar percent of sulfur element in polymer and the fourth column of “sulfonated AM unit (mol %)” is the molar percent of sulfonated acrylamide (AM) unit in the crosslinked PAM, whose calculation euqation has been also offered. We are so sorry for our previous error.

As for the composition of composite PAM-S(x)@PPy cryogels, our manuscript has the statement that “Since pyrrole was nearly polymerized into PPy, the amount of PPy in the composite cryogels kept the same”, which suggests the composition of PAM-S(x)@PPy cryogels.

- Figure 2 – has errors in representation.

Response: Thank you for pointing out this problem in our manuscript. We have corrected some error in Figure 2.

- the characterization of the compounds is presented without establishing a connection and interdependence between the methods.

Response: Thank you for this comment. We have confirmed the introduction of sulfonate groups into PAM cryogels by FTIR and elemental analysis, and also characterized the change of hydrophilicity based on the swelling behavior and dynamic contact angle of cryogels. As well, the change of pore structure was observed by SEM. After loading polypyrrole, FTIR, SEM and dynamic contact angle experiments were also used to characterize the structure and hydrophilicity changes, and the solar light absorption efficiency of the composite cryogels was also determined by UV-vis-NIR reflectance and transmittance spectroscopy. We hope that our expression offer sufficient information of the characterizations and their corresponding methods.

- FTIR spectra are not commented on the entire registered domain and without assigning the peaks corresponding to the present functional groups.

Response: Thank the reviewer for pointing out this problem. We have modified Figure 3 to show only the important local spectrum in the range 1900-900 cm-1 and more registered signals.

- For example, SR of PAM-S(0.5), the sample with the highest sulfonation degree, is about 17 g×g-1, which is significantly improved compared with 11 g×g-1 for pure PAM cryogel. – why? Explain

Response: It is an interesting comment. Because the sulfonate group is a more hydrophilic group and its hydrogen-bonding interaction with water is stronger, the sulfonated polymer chain is more stretched upon swelling and higher SR is reached..

- un-sulfonated = non-sulfonated.

Response: It is an interesting comment. We think that the un-sulfonated cryogel means that it has not been sulfonated, but it does not have sulfonate groups. Thus, we prefer to “un-sulfonated”.

- With the increase of sulfonation degree of PAM cryogel, the pore morphology even did not change obviously, but the pore diameter increased slightly (Figure 3g and 3h), which may be related to its swelling behavior. The larger pores of cryogels with higher swelling ratio (SR) were preserved after freeze-drying - the pore diameter depends on sulfonation degree, crosslinking degree?

Response: Thank for your comments. The crosslinking degree of cryogels before the sulfonation was the same, and the sulfonation experiment can not change the crosslinking degree, but the hydrophilicity of cryogels with different degrees of sulfonation will be different. The sulfonation degree might have a slight influence on pore size as the hydrophilicity increase would enlarge its pore upon contacting with water. As disclosed by the previous articles, the crosslinking degree affects the pore morphology of cryogels. However, our present study did not pay attention to it.

- The interaction between sulfonate group and pyrrole monomer helps the adsorption of pyrrole and the attachment of produced polypyrrole on the surface of PAM-S cryogel – authors must present the chemical adsorption of pyrrole onto the cryogel chemical surface and how this adsorption helps the polymerisation of pyrrole.

Response: Great thanks for the reviewer to point out such problem. Pyrrole is a weak base with secondary amino group while sulfonate group is an acid in HCl solution. Thus, the base-acid interaction as well as the possible hydrogen bonding help the adsorption [4, 5]. To avoid such the case in the future, we have added the related explanation in the revised manuscript.

- From SEM results (Figure 4g and h), it can be found that particles appeared on the surface of PAM-S(0.2)@PPy cryogel after loading polypyrrole. – which particles?

Response: We are appreciated with this exciting comment. They are PPy particles. During the oxidative polymerization of Py monomer absorbed by cryogel, some PPy particles were formed if the monomer was not attached on cryogel wall but just insides the cryogel pore.

in Supporting Information

- Composition comparison of sulfonated PAM before and after PPy loading – to clearly highlight the changes made in PAM-S(0.2)@PPy cryogels in fig. S2 the FTIR spectrum of PPy must be insert.

Response: Thank you for your comment. PPy is a common photothermal conversion material, and the typical IR spectral features of the PPy are also well reported in the attached references [6, 7].

- Figure S4. Thermogravimetric analysis results of PAM-S(0) (a), PAM-S(0.2) (b), PAM-S(0.2) (c), PAM-S(0.3) (d), PAM-S(0.5) (e) and PAM-S(0.2)@PPy (f) – what it means weight retention?

Response: Thanks for this interesting comment. The “weight retention” is referred to the weigh percent of the left weight of one sample upon heating it to certain temperature during TG analysis. TGA experiment determined the thermal stability of different cryogels to ensure the durability and reusability of the material.

  1. Zhao, F., X. Zhou, Y. Shi, X. Qian, M. Alexander, X. Zhao, S. Mendez, R. Yang, L. Qu, and G. Yu, Highly efficient solar vapour generation via hierarchically nanostructured gels. Nature Nanotechnology 2018. 13, 489-495.
  2. Li, H., M. Yang, A. Chu, H. Yang, J. Chen, Z. Yang, Y. Qian, and J. Fang, Sustainable Lignocellulose-Based Sponge Coated with Polypyrrole for Efficient Solar Steam Generation. ACS Applied Polymer Materials 2022. 4, 6572-6581.
  3. Wang, J.-Y., X.-X. Guo, J. Chen, S.-C. Hou, H.-J. Li, A. Haleem, S.-Q. Chen, and W.-D. He, A versatile platform of poly(acrylic acid) cryogel for highly efficient photothermal water evaporation. Mater. Adv. 2021. 2, 3088-3098.
  4. Müller, D., C.R. Rambo, D.O.S.Recouvreux, L.M. Porto, and G.M.O. Barra, Chemical in situ polymerization of polypyrrole on bacterial cellulose nanofibers. Synth. Met. 2011. 161, 106-111.
  5. Sahiner, N. and S. Demirci, The use of p(4-VP) cryogel as template for in situ preparation of p(An), p(Py), and p(Th) conductive polymer and their potential sensor applications. Synth. Met. 2017. 227, 11-20.
  6. Omastová, M., M. Trchová, J. Kovářová, and J. Stejskal, Synthesis and structural study of polypyrroles prepared in the presence of surfactants. Synth. Met. 2003. 138, 447-455.
  7. Blinova, N.V., J. Stejskal, M. Trchová, J. Prokeš, and M. Omastová, Polyaniline and polypyrrole: A comparative study of the preparation. Eur. Polym. J. 2007. 43, 2331-2341.

Besides the modification concerning with the reviewer comments, we have also made other changes to improve our manuscript. Those changes are also highlighted with blue color, some of them are listed belew.

1) Some changes of words color have been changed in Figure 2 to make them clearer.

To avoid the layout of Figure 6 across two pages, the first paragraph of Subsection 3.3 has been divided into two parts without the change of words. As well, Table 1 has been moved at the end of Subsection 3.4.

2) “(a)” has been added to the up-right SEM image of Figure S3. In the previous SI, it is lost.

Once again, we are very thankful for the great work by the editors and the comments from the reviewers. With those advices, our manuscript has been improved with our best efforts. Our revision and responses have addressed clearly the reviewing comments. We hope that our revised manuscript would be finally accepted by your honored journal.

Round 2

Reviewer 1 Report

The authors made a detailed explanations of the effects they experience during the study, which can be accepted. However, in my opinion, the authors believe that the questions arisen during reviewing are personal concerns of the reviewer and should not appear for other readers of the paper upon its publication. Thus, they made very little changes in the text, especially when compared with the detailed answers to the reviewers' queries.

With this, I disagree. I believe that, though no new concerns arise with the revised version, the second revision is needed, when these detailed answers from the rebuttal letter appear in the text. Otherwise, the text and discussion in the manuscript is not clear enough. I believe that this action will significantly improve the quality of discussion.

As minor issues

- new dotted lines in Figure 6 are invisible

- the explanation of incorrectness of replicates should be especially discussed in the text and reflected in figure captions because without the the explanation, such experiments seem to lack error bars, and it is authors responsibility to make it clear in the text

Author Response

Dear the editor and the reviewer:

We have made the concerning modification to our manuscript after careful thinking about the reviewing comments. The responses are listed below.

Comments and Suggestions for Authors

The authors made a detailed explanations of the effects they experience during the study, which can be accepted. However, in my opinion, the authors believe that the questions arisen during reviewing are personal concerns of the reviewer and should not appear for other readers of the paper upon its publication. Thus, they made very little changes in the text, especially when compared with the detailed answers to the reviewers' queries.

With this, I disagree. I believe that, though no new concerns arise with the revised version, the second revision is needed, when these detailed answers from the rebuttal letter appear in the text. Otherwise, the text and discussion in the manuscript is not clear enough. I believe that this action will significantly improve the quality of discussion.

Response: We strongly agree the viewpoint of the reviewer and thank once more for this suggestion. Since our previous responses are concerned with each other, and some of their relevant explanations served as the discussion of our manuscript. Thus, to avoid the repetition, we indeed made very little changes in the first revised manuscript. At the present version, we have done our best to improve our manuscript according to this suggestion from the reviewer and the changes are highlighted with red color.

As minor issues

- new dotted lines in Figure 6 are invisible

Response: We are appreciated with this comment. The following figures are Figure 6a in the initial and revised manuscript, respectively. The lines in the initial manuscript (left) are not exactly straight drawn by connecting two neighboring points, while those in the revised manuscript (right) is straight drawn from the fitting data points.

  See the attached word document.

- the explanation of incorrectness of replicates should be especially discussed in the text and reflected in figure captions because without the explanation, such experiments seem to lack error bars, and it is authors responsibility to make it clear in the text.

Response: We are appreciated with this comment and we also think that error analysis of data is very important. However, the measurements of water evaporation enthalpy and photothermal evaporation rate are done within certain duration and the slope of mass loss-time line is considered. As well, environmental temperature as well as humidity were not easy to modulate exactly. Thus, the following figures are Figure 6a in the initial and revised manuscript, the incorrectness of replicates in the articles of photothermal evaporation is usually not considered [Adv. Mater. 2021, 33, 2102994, DOI: 10.1002/adma.202102994; ACS Nano 2019, 13, 7913−7919, DOI: 10.1021/acsnano.9b02301; Nature Nanotechnology, 2018, 13, 489–495, https://doi.org/10.1038/s41565-018-0097-z].

Once again, we are very thankful for the great work by the editors and the comments from the reviewers. We hope that our revised manuscript would be finally accepted by your honored journal.

With best regards!

Sincerely yours,

Dr. Wei-Dong He (corresponding author), Email: [email protected]

Department of Polymer Science and Engineering

University of Science of Technology of China

Hefei, Anhui 230026, China
